# Type 2 Cystatins and Their Roles in the Regulation of Human Immune Response and Cancer Progression

**DOI:** 10.3390/cancers15225363

**Published:** 2023-11-10

**Authors:** Zijun Zhang, Fenghuang Zhan

**Affiliations:** Myeloma Center, Winthrop P. Rockefeller Cancer Institute, Department of Internal Medicine, University of Arkansas for Medical Sciences, Little Rock, AR 72205, USA; zzhang4@uams.edu

**Keywords:** cystatin, cathepsin, protease inhibitor, immunity, inflammation, tumor progression

## Abstract

**Simple Summary:**

Type 2 cystatins are a group of small secreted protease inhibitors that regulate cysteine protease cathepsins and legumain. These enzymes regulate important cellular processes that are linked to the immune response and tumor progression, playing important roles in both autoimmune diseases and various types of cancers. This review aims to explore the roles of type 2 cystatins in immune regulation and cancer development, shedding light on their significance in maintaining health.

**Abstract:**

Cystatins are a family of intracellular and extracellular protease inhibitors that inhibit cysteine cathepsins—a group of lysosomal cysteine proteases that participate in multiple biological processes, including protein degradation and post-translational cleavage. Cysteine cathepsins are associated with the development of autoimmune diseases, tumor progression, and metastasis. Cystatins are categorized into three subfamilies: type 1, type 2, and type 3. The type 2 cystatin subfamily is the largest, containing 10 members, and consists entirely of small secreted proteins. Although type 2 cystatins have many shared biological roles, each member differs in structure, post-translational modifications (e.g., glycosylation), and expression in different cell types. These distinctions allow the type 2 cystatins to have unique biological functions and properties. This review provides an overview of type 2 cystatins, including their biological similarities and differences, their regulatory effect on human immune responses, and their roles in tumor progression, immune evasion, and metastasis.

## 1. Introduction

Cysteine cathepsins, categorized as papain-like cysteine proteases, form a large group of cysteine-cleaving proteases that play crucial roles in various cellular processes, including extracellular matrix maintenance, immune surveillance, and cell infiltration [1]. The human cathepsin family comprises 11 members that are cysteine proteases. Cathepsin B, C, F, H, L, O, V, and X are ubiquitously expressed, whereas cathepsin K, S, and W are predominantly expressed in osteoclasts, cytotoxic lymphocytes, and antigen-presenting cells, respectively [2]. Cathepsins play important roles in oncogenesis and autoimmune diseases by mainly activating inflammatory responses and promoting extracellular matrix (ECM) degradation. Therefore, studying both cathepsins and their inhibitors holds immense value for understanding their role in both normal and diseased contexts.

Cystatins, originally isolated from chicken egg whites and subsequently found in various organisms, serve as natural inhibitors of cysteine cathepsins [3,4]. The human cystatin superfamily is categorized into three subfamilies (Figure 1A). Type 1 cystatins, also known as stefins, are small intracellular proteins of approximately 11 kDa that lack disulfide bonds. Type 2 cystatins, often called cystatins, are small (13–14 kDa) secreted proteins. Type 3 cystatins, also called kininogens, are large (88–114 kDa) glycosylated proteins that contain at least 1 type 2 cystatin-like cathepsin inhibitory sequence. Most cystatin members belong to the type 2 cystatin subfamily [5]. Although type 2 cystatins likely evolved from a common ancestral gene and share a similar overall structure, these cystatins exhibit distinct expression patterns, chemical properties, and biological functions. Therefore, it is likely that each member of the type 2 cystatin subfamily has different physiological processes within the human body. Since multiple cathepsins promote immune response activation, type 2 cystatins are natural immunosuppressants. Additionally, since cathepsins also enhance cell migration, type 2 cystatins also have anti-tumor functions. Type 2 cystatins also have non-protease inhibitory functions and can modulate several important immune and oncogenic pathways, making their role in tumor development and progression intricate. This review describes the immunomodulatory and oncologic roles of each type 2 cystatin with the goal of providing an overview of the immune-oncology axis of type 2 cystatins.

## 2. Cystatins as Protease Inhibitors

There are 10 known type 2 cystatin genes, each encoding a different cystatin protein (Table 1). All type 2 cystatin genes are located on chromosome 20p11.21, except for *CST6*, which is found on 11q13.1 [7]. Human type 2 cystatins share a common 3D structure (Figure 1B). The peptide sequence forms a protein with one beta sheet connected to an alpha helix connected to four beta sheets. This peptide folds into a structure in which the five beta sheets are arranged in an antiparallel form, covering the alpha helix at the top [5]. Several conserved regions are shared among human type 2 cystatins, and these conserved regions are highly relevant to their ability to inhibit protease function (Figure 2 and Appendix A). The G-QXVXG-VPW sequence is capable of inhibiting cathepsins and composed of three conserved sequences: the G near the N-terminus, the QXVXG fragment in the middle of the amino sequences, and the VPW fragment close to the C-terminus [5]. These three sequences form a wedge-shaped structure within the overall 3D structure and bind to the V-shaped cleft of cathepsin that contains the active site, thus effectively masking the active site and inhibiting protease function [5,8]. Additionally, three members of type 2 cystatins—cystatin C, E/M, and F—can inhibit asparaginyl endopeptidase (i.e., legumain), one of only two members of the peptidase family C13 that is closely related to the cathepsin superfamily C1 [4,9]. The conserved post-helix N residue in type 2 cystatins directly binds to the active site of legumain, which is located on the opposite side of the cathepsin inhibitory site in the 3D structure [4,10,11,12]. Thus, cystatin C, E/M, and F act as competitive inhibitors of legumain. The following subsections will briefly cover the features and functions of each type 2 cystatin in healthy individuals.

### 2.1. SD-Type Cystatins: Cystatin SN, SA, S, D (CST1, CST2, CST4, CST5)

The SD-type cystatins, named from the combination of cystatin S and cystatin D, include four members: cystatin SN, SA, S, and D. Cystatin SN, SA, and S contain 113 amino acids and are approximately 90% identical to each other. Along with cystatin D, these proteins share an identity of more than 55% [17,18,19]. These cystatins are exclusively expressed in parotid and submandibular glands and are found in human saliva, hence why they are sometimes called “salivary cystatins” [13]. The SD-type cystatins play an important role in maintaining a healthy oral environment. SD-type cystatins inhibit proteolytic events induced by cathepsin B, H, and L. The dysregulation of these proteolytic events causes periodontal tissue destruction which ultimately leads to bacteria colonization and periodontal disease [20]. Additionally, SD-type cystatins show anti-microbial function and can inhibit the growth of multiple bacteria strains, including *Aggregatibacter actinomycetemcomitans*, *Streptococcus pyogenes*, and *Porphyromonas gingivalis* in in vitro experiments [21,22,23]. The anti-microbial mechanism of SD-type cystatins is not fully understood, but SD-type cystatins may be inhibiting proteolytic activity necessary for growth promotion in the bacteria [24]. This antimicrobial effect is also observed in human cystatin C and cystatins isolated from other species [25]. Given the conserved nature of this antimicrobial effect, type 2 cystatins may have originally served as antimicrobials [26].

Cystatin SN (CYS1), encoded by the gene *CST1*, is the most studied SD-type cystatin. CYS1 has a similar cathepsin inhibitory function with the ubiquitous cystatin C; however, the inhibitory efficiency of CYS1 is much lower than that of cystatin C [8,19,27].

Cystatin SA, also referred to as cystatin 2 (CYS2), is encoded by *CST2*. Its amino acid sequence is 87% identical with cystatin SN and 90% identical with cystatin S. The *CST2* locus has two allele variants that encode cystatin SA1 and cystatin SA2. These alleles only differ by two amino acids (SA2 contains G79D and E140D polymorphisms in comparison to SA1). The gene frequency of cystatin SA1 is 0.935, and the gene frequency of cystatin SA2 is 0.065 [28]. The G79D polymorphism is within the highly conserved QXVXG fragment, and this makes cystatin SA2 a weaker protease inhibitor than cystatin SA1 [29]. Currently, there are no studies that reveal any other function of cystatin SA2.

Cystatin S (CYSS), originally called SAP-1, is encoded by *CST4* [30,31]. CYSS shows very similar activity to cystatin SN and SA, but CYSS contains a higher Asp+Glu/Asn+Gln ratio, making CYSS a more acidic protein [32]. CYSS has an F instead of a V within the QXVXG fragment. This makes CYSS the weakest cathepsin inhibitor among all the type 2 cystatins [19]. CYSS is largely expressed in the submandibular and parotid glands of humans and rats [33]. Although it is a weak cathepsin inhibitor, CYSS retains an antibacterial function and inhibits the growth of *Porphyromonas gingivalis* [23]. Since its discovery, CYSS is most often used as a prognostic biomarker of dry eyes [34].

Cystatin D (CYSD) is encoded by *CST5*, a homolog of *CST1*, *CST2*, *CST3*, and *CST4.* CYSD is a small 122-amino-acid protein that shares more than 50% identity with cystatin SA, SN, C, and S. *CST5* has two alleles, which encode two forms of CYSD. The two forms of CYSD differ by either a C or an R at residue 26. Both variants act as protease inhibitors and have the same activity [35]. Compared with other SD-type cystatins, CYSD is a less potent inhibitor of cathepsin L and does not inhibit cathepsin B. However, CYSD is a much stronger inhibitor of cathepsin S and cathepsin H [18]. Clinically, *CST5* is considered an early biomarker of traumatic brain injury [36].

### 2.2. Cystatin C (CST3)

*CST3* encodes cystatin C (CYSC), which was first found in human saliva [37]. *CST3* was originally categorized as an SD-type cystatin until scientists realized that *CST3* expression is not limited to parotid and submandibular glands [38]. Instead, it is ubiquitously expressed in all nucleated cells [14,39]. *CST3* is likely the ancestral gene of all the members of the type 2 cystatin family [19]. *CST3* is a housekeeping gene. It is ubiquitously expressed at relatively low levels throughout the human body [14]. The highest *CST3* expression is found in the brain and salivary glands. CYSC is also present in all body fluids, with the highest CYSC concentration found in brain fluid, semen, and breast milk [14]. This indicates that CYSC may be related to reproduction. However, its reproduction-related function and mechanism remain to be elucidated. Because of its small size, CYSC can be filtered by the glomerulus without being reabsorbed by the kidney’s proximal convoluted tubule. At the same time, the proximal tubular cells catabolize their own CYSC but do not secrete it [40]. As a result, CYSC is widely used as a marker for the estimated glomerular filtration rate (eGFR), a standard for kidney function diagnosis [41,42].

CYSC can dimerize by 3D domain swapping, a process in which two CYSC molecules exchange the position of their α1 helix and β1 and β2 strands [43]. During this exchange, the L1 loop of CYSC, which is required for cathepsin inhibitory activity, is disturbed. As a result, dimerized CYSC is inactive and unable to inhibit cathepsins [43,44]. Extracellular CYSC is mainly produced by macrophages and dendritic cells (DCs). Cells synthesize CYSC as both dimers and monomers; however, only the active monomer is secreted into the extracellular space [44].

Like the SD-type cystatins, CYSC also has antibacterial activity [22]. Since CYSC is a housekeeping protein, it is likely one of the systematic regulators of general cellular cathepsin activity. In the brain, cathepsin B, one of the known targets of CYSC, degrades amyloid beta, a protein associated with neurodegeneration [45,46]. At the same time, the domain-swapping property of CYSC can also promote amyloid beta plaque formation. CYSC dysregulation induced by a polymorphism or mutation is highly linked to the pathogenesis of neurodegeneration and Alzheimer’s disease [45,47,48]. A study conducted by Wahlbom et al. showed that CYSC with L86Q, a common mutation in cerebral amyloidosis patients, does not form normal dimers by domain swapping. Instead, CYSC oligomerizes by continuously inserting its α1-β1-β2 domain into the flexible region of the next CYSC molecule. This leads to the formation of a large donut-shaped oligomer, which eventually causes amyloid fibril generation [49].

### 2.3. Cystatin E/M (CST6)

Cystatin E/M (CYS6) was simultaneously identified by two groups: one group isolated CYS6 from epithelial cells and named it cystatin E; the other group identified CYS6 mRNA downregulated in breast cancer, and they named it cystatin M [50,51]. Eventually, scientists discovered that the same gene encodes cystatin E and M and the proteins have the same amino acid sequence. Therefore, the protein encoded by *CST6* was renamed cystatin E/M. Compared with other members of the type 2 cystatin subfamily, *CST6* is the only cystatin gene located on chromosome 11q13 [52]. CYS6 is one of three cystatins that can inhibit cathepsins and legumain and is currently the strongest legumain inhibitor. CYS6 is one of two glycoproteins in the type 2 cystatins subfamily. It has an N-linked glycosylation site at residue 137, attached to a mannose-6-phosphate-rich glycan [53]. The function of the glycan on CYS6 has not been deduced in the current literature. Similar to CYSC, CYS6 can dimerize via domain swapping; however, the CYS6 dimer is only generated in vitro in a high-temperature, destabilizing condition [54].

In healthy individuals, low levels of CYS6 were found in most human tissues. High levels of CYS6 were found in human cutaneous epithelia, hair follicles, sebaceous glands, and sweat glands [15]. Within the category of mucosal body fluids, the highest concentration of cystatin E/M was found in semen, with levels as high as 500 ng/mL [55]. This indicates that CYS6 may play a role in reproduction, but currently, there are no studies in this area. High cystatin E/M expression in epidermal tissue is highly related to skin development and epidermal homeostasis. One of the known targets of CYS6, cathepsin V, degrades desmosomal proteins, including DSG1, DSC1, and CDSN. Another target of CYS6, cathepsin L, activates transglutaminase 3 [56]. Cystatin E/M regulates the desquamation and cornification process by inhibiting cathepsin V and L activity with high efficiency. The dysfunction of *CST6* results in dry skin and keratosis caused by unregulated proteolytic events induced by cathepsin V, L, and legumain in the epidermis [57,58]. Knockdown of *CST6* in vitro causes deficient development of the multilayer epidermis [59]. Mice null for *CST6* (i.e., ichq mice) exhibit multiple abnormalities related to epidermal development, including keratosis, dry skin, hair loss, hypotrichosis, and even neonatal lethality. Fortunately, these phenotypes can be partially rescued by cathepsin L knockdown [60].

### 2.4. Cystatin F (CST7)

Cystatin F (CYSF), also referred to as cystatin-like metastasis-associated protein (CMAP) or leukocystatin, is encoded by *CST7* [16,61]. Human CYSF is a glycoprotein that has 145 amino acids. It has a glycosylation site on N62 and N115 [62]. It shares approximately 35% identity with CYSC, 30% with SD-type cystatins, and 32% with CYS6 [62]. CYSF is a potent inhibitor of cathepsin F, K, and V and it can also weakly inhibit cathepsin S and H. CYSF cannot inhibit cathepsin C and X in kinetic measurement experiments when only the enzyme and the substrate are present in the system [63]. However, the inhibition of cathepsin C can be observed in myeloid cells in both in vitro and in vivo experiments [64,65,66]. This activity difference is attributed to the cellular processing of CYSF, where cleavage of the N-terminus region enables CYSF to inhibit cathepsin C. CYSF is selectively expressed by hematopoietic cells, especially in NK cells, T cells, and DCs [16,62]. Tissue from the spleen, bone marrow, lymph nodes, and lung express the highest levels of *CST7* [14,67].

CYSF is capable of dimerization through disulfide bonds, which differs from the dimerization mechanism of domain swapping utilized by CYSC and CYS6. CYSF is synthesized as a dimer, which is inactive due to steric hindrance [68]. After synthesis, the inactive dimer is delivered to the lysosome and cleaved by the proteasome, a process mediated by the glycosylation on CYSF [68]. Similar to all type 2 cystatins, CYSF is a secreted protein; however, after secretion, CYSF is quickly taken up by its secreting cell and delivered to the lysosome as a functional protease inhibitor via the mannose-6-phosphate pathway [68]. CYSF is an important immune modulator in healthy individuals. The role of CYSF as an immune modulator is covered in detail in the next section.

### 2.5. Testatins: CST8, CST9, CST11

Cystatin 8, 9, and 11 are selectively expressed in the testis and are thought to play a role in the reproductive system [69]. Therefore, they are also referred to as testatins. Currently, there are very few papers studying testatins and no detailed conclusion could be drawn on their accurate function. One paper reported that the loss of function of CST8 is related to reduced fertility and abnormal testis development [70]. Although a few studies have reported that CST9 is linked to cancer development, no study has focused on the oncogenic roles or immune modulatory roles of testatins [71]. Therefore, they will not be covered in this review.

## 3. Cathepsin, Legumain, and Cystatins as Immune Modulators

To introduce the immune-modulatory role of cystatins, it is important to also briefly cover the immunomodulatory role of their target proteins, which are cathepsins and legumain. Cathepsin C, S, L, K, and legumain are involved in autoimmune regulation and exhibit pro-inflammatory functions.

Cathepsin C regulates the maturation of neutrophil serine proteases (NSPs). The secretion of NSPs helps to degrade invading microorganisms and represents one of the major anti-bacterial mechanisms of neutrophils. NSPs also participate in inflammatory response regulation by proteolytically modifying or degrading extracellular inflammatory cytokines and chemokines and catalyzing the activation of inflammatory response-related receptors [72]. NSPs are initially generated as zymogens, which have a dipeptide structure at their N-terminus, keeping the protein in an inactive state. Cathepsin C cleaves this dipeptide to activate the NSPs [73]. In addition, cathepsin C regulates the activity of granzyme B, which is related to the immune response of CD8+ T cells and natural killer (NK) cells [74].

Cathepsin S is required for MHC class II antigen presentation since maturation of the MHC class II molecules depends on cathepsin S activity [75,76]. In antigen-presenting cells, the MHC-li chaperone complex is first generated. The li peptide in the MHC-li complex is a 10 kDa invariant peptide chain that masks the antigen binding site of MHC class II molecules, thereby inactivating the whole MHC class II molecule. The MHC-li complex is then delivered to the lysosome and the li peptide is cleaved by cathepsin S to activate the MHC class II molecule [77,78]. As a result, cathepsin S is mainly an immune modulator that enhances immune surveillance.

Cathepsin L is also involved in the maturation of MHC class II molecules in macrophages [79]. Additionally, cathepsin L is an activator of perforin, a key enzyme involved in pore formation by cytotoxic lymphocytes [74,80].

Cathepsin K, a collagenase that is selectively expressed in osteoclasts, is associated with autoimmune disease. It plays an important role in bone matrix remodeling and mediates the inflammatory stress on the bone surface by degrading collagen I, II, and elastin [75,81].

Legumain, as the target of CYSC, CYS6, and CYSF, also participates in immune regulation. Like cathepsin S and L, legumain promotes the maturation of MHC class II molecules and participates in the direct cleavage of antigens, thereby facilitating overall antigen presentation. Legumain can also activate Toll-like receptors, an upstream receptor of the Toll-like receptor signaling pathway, which are key players in the innate immune response [82].

Overall, cathepsins and legumain can trigger immune responses, and their dysregulation often correlates with the pathogenesis of autoimmune diseases, including rheumatic arthritis, systemic lupus, Sjogren’s syndrome, asthma, and psoriasis [75].

Since type 2 cystatins inhibit cathepsins and legumain, which in turn suppress inflammatory responses and immune cell activation, type 2 cystatins can be considered immune modulators. Indeed, most of the type 2 cystatins are immunosuppressive due, at least in part, to their ability to inhibit protease function. Type 2 cystatins play key roles in DC maturation and neutralizing cytotoxic lymphocyte cytotoxicity. Additionally, some of them can also act as ligands to activate anti-inflammatory pathways. The following subsections will talk about this in detail for each member of type 2 cystatins.

### 3.1. Cystatin SN (CYS1)

CYS1 contributes to allergic responses and the pathogenesis of allergic respiratory diseases. CYS1 and cystatin SA are highly expressed in the nasal epithelial surface of patients who have chronic rhinosinusitis (CRS) with nasal polyps. Using proteomics, researchers have discovered that the mucus of CRS patients contains high concentrations of CYS1 [83,84]. CYS1 is related to the Th2 immune response—an inflammation-related response that is highly associated with allergic responses and induced by IL-4. Currently, CYS1 serves as a biomarker for the activated Th2 immune response, which is mainly triggered by CD4+ T cells [85]. *CST1* expression can be induced by IL-4 stimulation and is positively correlated with the upregulation of the Th2 immune response markers IL-33 and TSLP in eosinophilic CRS cases [86]. In addition, recombinant human CYS1 induces secretion of several Th2-related cytokines, including IL-5, IL-13, and IL-4, with an increase in Th2 cell infiltration [83]. For this reason, *CST1* is a well-accepted biomarker for CRS diagnosis. Since CYS1 regulates the Th2 immune response, it is a good biomarker for asthma and allergic rhinitis. *CST1* is upregulated in epithelial cells of the upper and lower airways in patients with asthma and allergic rhinitis [87,88]. The upregulation of *CST1* is correlated with severe allergic respiratory disease [89,90]. Interestingly, allergic respiratory diseases are treated with corticosteroids which downregulate *CST1* expression in the airway epithelial cells [87]. One study showed that CYS1 promotes the proliferation and migration of airway smooth muscle cells by activating the PI3K/AKT signaling pathway in an in vitro asthma model. This indicates that CYS1 participates in airway remodeling events and maybe a main contributor to asthma development [91].

Because CYS1 promotes the secretion of IL-4 in the immune cells, CYS1 may introduce an anti-inflammatory response in the microenvironment. Indeed, patients with CRS who have high *CST1* expression in their nasal epithelial cells also have high levels of CCL18. CCL18 is an M2 macrophage polarizing cytokine that induces immunosuppression and anti-inflammatory responses [92,93]. In addition, *CST1* plays an important role in introducing an anti-inflammatory response in acute liver failure (ALF) models [94]. CYS1 can directly bind to IFNGR1/2, acting as a competitive inhibitor of the pro-inflammatory cytokine IFN-γ. As a consequence, CYS1 inhibits the activation of the JAK/STAT1 pathway induced by IFN-γ and promotes the M2 polarization of macrophages in the liver [94]. In summary, CYS1 plays a role in the allergic Th2 immune response and anti-inflammatory processes.

### 3.2. Other SD-Type Cystatins (CST2, CST4, CST5)

In addition to *CST1*, clinical studies have shown that the other SD-type cystatins (*CST2*, *CST4*, and *CST5*) also participate in immune response regulation. To date, the mechanism behind their immune regulatory function is unknown. As a result, conclusions about their function in immune regulation cannot be drawn.

Although *CST1* is the preferred clinical biomarker for allergic respiratory diseases, like asthma and CRS, *CST2* can also be used as a clinical biomarker for allergic respiratory diseases [95,96,97]. *CST4* is used as an anti-inflammatory marker since it is downregulated in rheumatic arthritis and Sjogren’s syndrome [98,99].

Unlike the other SD-type cystatins which are used as anti-inflammatory biomarkers, *CST5* is used as an inflammatory biomarker [100,101,102]. An in vitro experiment on the human MRC-5 diploid lung cell transfection model showed that CYSD inhibits the replication of OC43 and 229e coronavirus strains at its physiological concentration, showing that CYSD has an anti-viral function [103].

### 3.3. Cystatin C (CYSC)

CYSC has been shown to have anti-inflammatory properties. Immature DCs express high levels of *CST3*, and this upregulation gradually disappears during DC maturation [104]. The expression of CYSC protein in immature DCs prevents the protease activity of cathepsin S from activating the MHC class II molecules. Therefore, immature DCs are less able to complete antigen presentation and a following immune response [105]. A high level of CYSC positively correlates with the severity of several autoimmune diseases, including sepsis and rheumatoid arthritis [106,107]. A high level of CYSC also correlates with more severe HIV infections, which can be suppressed by antiviral treatment [108].

On the other hand, CYSC also shows pro-inflammatory activities. In hematopoietic cells, CYSC is expressed at higher levels in macrophages and DCs than in T cells [44,109]. In macrophages, CYSC modulates the immune response by enhancing macrophage responses to IFN-γ. This consequently upregulates the activation of the NF-κB pathway and promotes immune-related cytokine NO and TNF-α secretion [110]. CYSC expression can inhibit the secretion of the anti-inflammatory cytokine IL-10 and it is an antagonist of TGF-β [111,112]. At the same time, IL-10 and TGF-β can also modulate the expression of CYSC via the upregulation of its transcription factor IRF-8, forming a negative feedback loop [110,111].

### 3.4. Cystatin E/M (CYS6)

The immunomodulatory role of CYS6 has not been reported in the current literature. However, some predictions can be made based on the current mechanistic studies of CYS6. CYS6 can inhibit the activation of NF-κB signaling by both canonical and non-canonical pathways [113,114]. CYS6 can also be internalized by macrophages where it highly suppresses osteoclast cathepsin K [114]. Given this evidence, CYS6 has the potential for development as a therapeutic agent to alleviate rheumatic arthritis [75].

### 3.5. Cystatin F (CYSF)

CYSF is an important immune modulator that acts as a natural immunosuppressant in the human body. High levels of CYSF in NK cells, T cells, and DC cells have been used to counteract the activated immune response mediated by cathepsins. A high level of CYSF reduces the cytotoxicity of CD8+ T cells and NK cells. The inhibition of cathepsin C, L, and H in NK cells and CD8+ T cells suppresses the expression of the downstream cytotoxic-related protein perforin and granzymes [115,116]. In contrast, CYSF co-localizes with cathepsin S in immature DCs and weakens as the DCs mature [117]. The inhibition of cathepsin S by CYSF in immature DCs likely suppresses antigen presentation induced by the MHC class II molecules in DCs. Studies suggest that CYSF is associated with inflammatory responses in the central nervous system induced by microglia, a type of cell that is closely related to macrophages in the neural system [118,119].

In conclusion, besides CYSD and CYSC’s non-protease inhibitory function, the type 2 cystatins are all anti-inflammatory factors and are potential immunosuppressants.

## 4. The Role of Cathepsins, Legumain, and Type 2 Cystatins in Cancer Development

Type 2 cystatins are cysteine cathepsin inhibitors and this is also the reason why they have immunosuppressive effects. Theoretically, type 2 cystatins should have pro-tumor effects on cancer models as they are immunosuppressants. However, the situation is complex because several cathepsins also play roles in tumorigenesis-enhancing cellular events. The activity of cathepsin B, a ubiquitous cathepsin, is highly related to tumor progression and various devastating immunosuppressive events. These events include reduction in CD8+ T cell persistence, infiltration of immunosuppressive tumor-associated macrophages (TAMs), myeloid-derived suppressor cells (MDSCs), and regulatory T cells (Tregs) [75,120]. In addition, cathepsin B, L, S, and K participate in the cleavage and degradation of the extracellular matrix (ECM). As a result, the activation of these cathepsins facilitates the migration, invasion, and metastasis of tumor cells [121]. Cathepsins also have a protective effect on tumor cells and are involved in the chemoresistance of cancer.

Legumain is largely expressed on TAMs and cancer cells from the breast, prostate, and liver. The overexpression of legumain is highly correlated with tumor migration, invasion, poor prognosis, and inferior survival. Legumain can activate MMP-2&9, PI3K/AKT, and integrin signaling pathways to promote epithelial-mesenchymal transition (EMT) and the TGF-β signaling pathway. Additionally, it also cleaves and inactivates the tumor suppressor protein p53 [122].

Since type 2 cystatins inhibit both protease-induced inflammatory responses and pro-tumor activity, they exhibit a complex dual effect on tumorigenesis. In the following subsections, the role of each type 2 cystatin in cancer development is discussed.

### 4.1. Cystatin SN (CYS1)

*CST1* is a pro-tumor gene in multiple cancer models, including esophageal, breast, colon, gastric, liver, and lung cancer [123,124,125,126,127,128]. In lung cancer, *CST1* is hypomethylated [129], resulting in upregulation of *CST1*. This upregulation is often correlated with poor prognosis, metastasis, and recurrence [124,125,127,128]. Compared to CYSC, CYS1 is a weaker inhibitor but it displays a higher affinity for cathepsin B, which is known to be involved in tumor progression. Therefore, overexpression of CYS1 can partially neutralize the inhibitory effect of CYSC on cathepsin B, inducing stronger tumor invasiveness mediated by enhanced cathepsin B activity [130]. In addition to its ability to neutralize CYSC, CYS1 also upregulates AKT phosphorylation and activates the PI3K/AKT pathway. This subsequently downregulates E-cadherin and facilitates EMT, promoting tumor metastasis [128]. CYS1 directly interacts with the ferroptosis mediator GPX4 by interfering with the ubiquitination of GPX4 and stabilizing it, thus inhibiting tumor cell ferroptosis. This consequently promotes tumor progression and metastasis in gastric cancer models [123].

### 4.2. Other SD-Type Cystatins (CST2, CST4, CST5)

Since SD-type cystatins share a highly similar amino acid sequence and protein structure, it is not surprising that CYS2 and CYSS exhibit pro-tumor functions. However, studies on these cystatins are limited, which makes it difficult to conclude whether they have the same pro-tumor mechanism as CYS1. Upregulation of *CST2* correlates with tumor metastasis of prostate and gastric cancer [131,132]. CYS2 may help regulate the TGF-β signaling pathway and promote EMT like CYS1, but the mechanism needs further exploration to be certain [131]. *CST4* is upregulated in esophageal, colon, and gastric cancer and is associated with poor prognosis [133,134,135]. The overexpression of CYSS in gastric tumor cells upregulates a protein called ELFN2, which directly inhibits the expression of E-cadherin and promotes EMT, like CYS1 and CYS2 [134]. However, there are very few studies on ELFN2 and its connection with tumor invasiveness. Thus, future studies are needed to determine how CYSS promotes tumor growth.

In contrast, based on biomarker studies of colon, gastric, and prostate cancer, *CST5* is considered an antitumor gene [136,137,138]. *CST5* expression is induced by vitamin D, which is known to have antitumor activity and is downregulated in human colon cancer cells [139,140,141]. CYSD inhibits the Wnt/β-catenin signaling pathway and oncogenic c-MYC expression, which consequently extends the cell cycle and reduces the proliferation, migration, and invasiveness of colorectal tumor cells [139]. Importantly, some studies have suggested that *CST5* is a downstream target of p53 [142]. The p53 protein enhances *CST5* expression by interacting with the *CST5* promoter region and simultaneously downregulating EMT-promoting transcription factor SNAIL, which in turn downregulates *CST5* expression [142].

### 4.3. Cystatin C (CYSC)

CYSC has a dual effect on cancer development. It was observed to have an antitumor function on several cancer types, including pancreatic cancer, breast cancer, and leukemia [143,144,145,146]. Currently, there are at least two mechanisms that can explain the antitumor function of CYSC. First, the antitumor effect of CYSC is highly related to its strong inhibitory effect on cathepsin B. Compared with other type 2 cystatins, CYSC is the strongest inhibitor of cathepsin B [147]. Because CYSC inhibits cathepsin B, CYSC acts as a suppressor of cell migration, thereby inhibiting tumor invasion and metastasis [148]. CYSC is also an inhibitor of the TGF-β pathway. CYSC directly interacts with TGF-β receptor 2 and competes with the binding of the TGF-β ligand. By inhibiting the TGF- β signaling pathway, CYSC inhibits multiple metastatic events in tumor cells, such as loss of cell contacts, downregulation of cell polarization, and increased cell migration [149].

In contrast, there are also several studies stating that a high CYSC level correlates with a worse prognosis. High levels of CYSC were found in tumor tissue of ovary, colon, and esophageal cancer [150,151,152]. Cathepsin B has been found to promote apoptosis and this could be linked to the observed phenomenon that CYSC suppresses apoptosis of tumor cells [153,154]. However, the mechanism by which CYSC promotes tumor progression is unknown. CYSC can be used as a biomarker for cancer prognosis. However, recent studies suggest that renal function, which is frequently altered in cancer, heavily influences the levels of serum CYSC and intercellular CYSC [155,156]. This could negatively impact the use of CYSC as an effective biomarker for cancer prognosis.

### 4.4. Cystatin E/M (CYS6)

CYS6 has been found to have dual effects, exhibiting pro-tumor properties in some cancers and anti-tumor properties in others. *CST6* is an antitumor gene that is hypermethylated in breast, prostate, brain, and cervical cancer [113,157,158,159]. The antitumor effect of *CST6* has been mostly studied in breast cancer models and its downregulation correlates with higher levels of migration and invasiveness in both in vivo and in vitro models of breast cancer [157]. The epigenetic silencing of *CST6* is strongly correlated to breast cancer bone metastasis [157]. CYS6 also inhibits the pro-oncogenic function of legumain [160,161]. In addition, CYS6 inhibits cathepsin B, which is frequently upregulated in breast cancer [162]. Cathepsin B cleaves SPHK1, an enzyme that inhibits osteoclast differentiation by inhibiting p38 activation induced by RANKL [162]. Uncontrolled osteoclast differentiation disrupts bone homeostasis, causes bone disease, and promotes cancer bone metastasis [163]. Furthermore, CYS6 inhibits cathepsin K, an enzyme that is predominantly expressed in osteoclasts and participates in bone matrix remodeling [114]. Therefore, CYS6 suppresses bone metastasis of tumor cells by inhibiting osteoclastogenesis. CYS6 also inhibits NF-κB signaling in both canonical and non-canonical pathways [113,114].

In contrast, *CST6* has also been found to have pro-tumor functions in pancreatic, liver, gastric, and triple-negative breast cancer [164,165,166,167]. Additionally, a small fraction of patients with multiple myeloma express high levels of *CST6* [114]. A recent paper has identified *CST6* as a factor involved in the dysregulation of necroptosis in gastric tumor cells, resulting in a poor prognosis for patients with gastric cancer [167]. Compared to our understanding of the antitumor effects of CYS6, the mechanism by which CYS6 promotes tumor growth is poorly understood.

### 4.5. Cystatin F (CYSF)

Although there are extensive reports on the immunoregulatory function of CYSF, the role of CYSF in cancer is unclear. Given the immunosuppressive function of CYSF, uncontrolled upregulation of *CST7* likely enhances tumor progression. Indeed, high levels of *CST7* are associated with poor prognosis and lower survival rates in patients with liver, oral, and brain cancer [116,168,169]. The current literature has shown that the pro-tumor effects of CYSF are highly correlated to its immunosuppressive function. Upregulation of *CST7* causes a decrease in the cytotoxicity of NK cells to tumor cells [168]. In contrast, multiple studies suggest that *CST7* downregulation is associated with higher invasiveness of tumors, more metastatic events, and tumor progression in prostate cancer, lung cancer, pancreatic cancer, and lymphoma [170,171,172,173]. Since *CST7* downregulation is correlated with metastasis and tumor progression, this indicates that *CST7*, like other type 2 cystatin family members, regulates the cathepsin B-L metastatic axis. However, further studies are needed to draw definitive conclusions on the role of *CST7* in cancer progression.

## 5. Type 2 Cystatin Studies on Mouse Model: A Brief Overview

Although type 2 cystatins play important roles in human immune responses and tumor progression, it is important to choose a good animal model for cystatin in vivo studies. Fortunately, the sequence homology and function of many type 2 cystatins are highly conserved in mice. *CST3*, *CST5*, *CST6*, and *CST7* are highly conserved in mice, with very close similarities in their protein sequences compared to humans (Appendix A) [174]. According to the current literature, mouse models, especially the C57/BL strain (the gene of interest could be manipulated) are frequently used in *Cst3*, *Cst6*, and *Cst7* studies and the results harmonize with the clinical data obtained in humans [162,175,176]. Although mice lack matched genes for *CST1*, *CST2*, and *CST4*, studies show that human recombinant CYS1 and CYS2, which are the matched proteins of *CST1* and *CST2*, are also functional in mouse models and their findings align with clinical data obtained from humans [83,123]. These studies provide evidence that the function of many type 2 cystatins is also highly conserved in mouse models and can serve to uncover their unique functions at the immune-oncology axis.

## 6. Conclusions

Type 2 cystatins are the natural inhibitors for a group of proteases called cysteine cathepsins. Since their discovery, type 2 cystatins have emerged as valuable biomarkers for various autoimmune-related diseases and cancer. These small secreted protease inhibitors suppress proteolytic activities that are necessary for immune cell development and polarization, protein maturation, and cytotoxicity exertion. Therefore, type 2 cystatins are potential immunosuppressants. Secondly, they inhibit the proteolytic activity necessary to promote tumor metastasis, such as ECM degradation and EMT. Finally, the regulatory role of type 2 cystatins extends beyond protease inhibition, since they can interact with receptors involved in crucial immune-oncogenic pathways (Figure 3 and Table 2). This suggests the presence of additional structural regions within type 2 cystatins that may function as ligands for these receptors. There are conserved regions in type 2 cystatins with unknown functions, and these regions may play a role in ligand-receptor interactions (Figure 2 and Appendix A). Thus, it might be worthwhile to re-evaluate the structure of type 2 cystatins, as it potentially provides insights into their non-protease inhibitory functions.

Despite the frequent use of type 2 cystatins as biomarkers, few studies have investigated the mechanisms underlying their immune regulatory functions and their pro-tumor and antitumor effects on tumor development. Given the immense potential of type 2 cystatins as novel immunosuppressants, anticancer agents, and targets for cancer therapeutics, future studies should investigate the molecular and cellular mechanisms of type 2 cystatins. Such investigations should not only focus on their protease inhibitory function but also delve into the characteristics that go beyond protease inhibition.

## Figures and Tables

**Figure 1 cancers-15-05363-f001:**
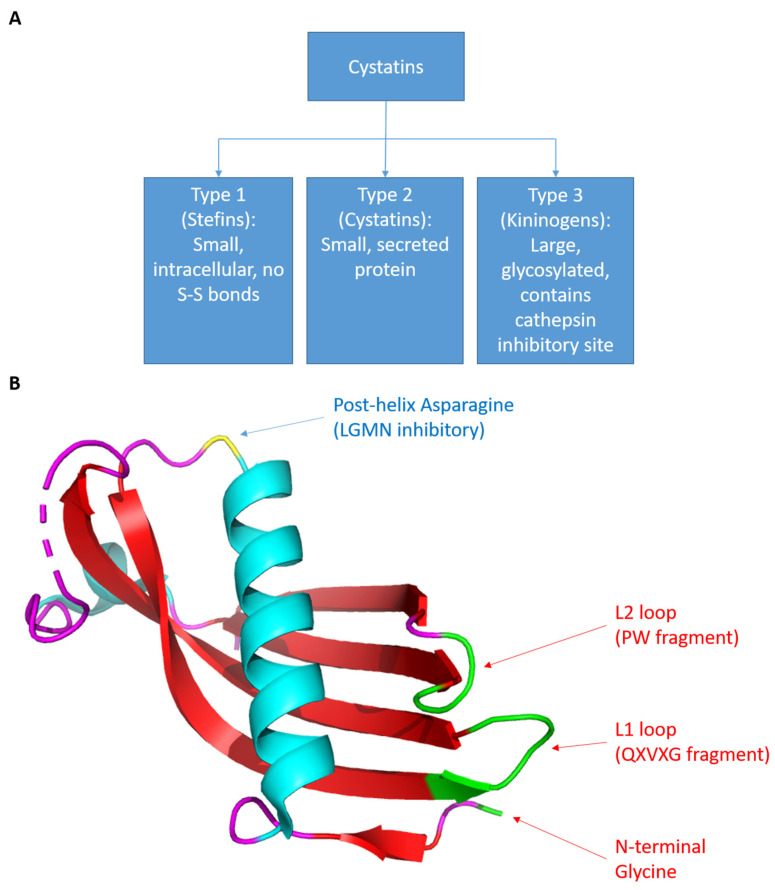
(**A**) The categorization of cystatins. (**B**) The structure of cystatins. Using cystatin C as an example. The coordinate file was retrieved from RCSB PDB, id: 3GAX. The image was generated using PyMOL (Schrodinger). The green region represents the cathepsin inhibitory site and the yellow residue represents the LGMN inhibitory site. This structure was originally discovered and published by Kolodziejczyk et al. [6].

**Figure 2 cancers-15-05363-f002:**
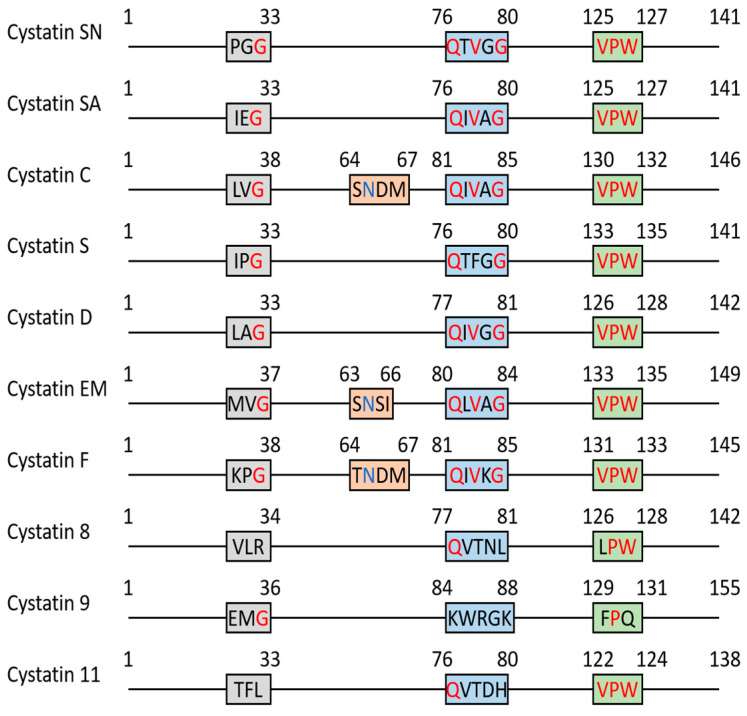
The schematic diagram of key region alignment of type 2 cystatin family. The sequence of each cystatin was retrieved from the NCBI GenPept database. Multiple sequence alignment was performed by Clustal Omega and Boxshade. The conserved G-QXVXG-VPW fragment is highlighted in red and post-helix N is highlighted in blue.

**Figure 3 cancers-15-05363-f003:**
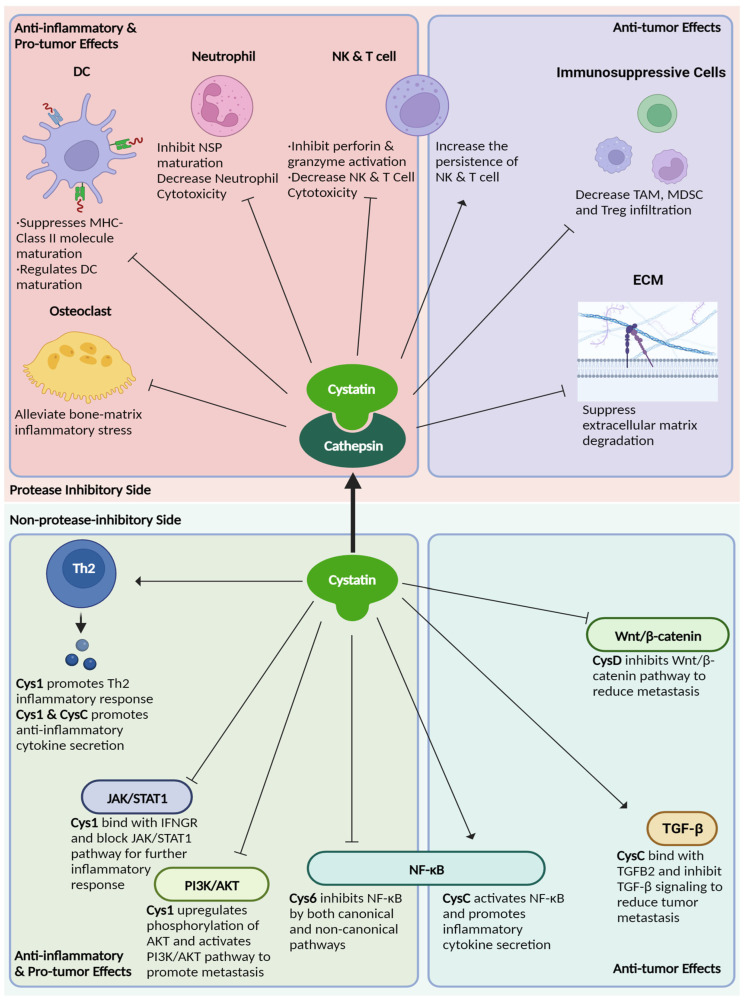
The overview of type 2 cystatins’ role on immune regulation and cancer development.

**Table 1 cancers-15-05363-t001:** List of Type 2 Cystatins. Gene location info retrieved from the NCBI database of each gene.

Name of Gene	Encoded Protein	Gene Location	Site of Expression
*CST1*	Cystatin SN	20p11.21	Parotid gland, lacrimal gland [13]
*CST2*	Cystatin SA	20p11.21	Parotid gland [13]
*CST3*	Cystatin C	20p11.21	Ubiquitous [14]
*CST4*	Cystatin S	20p11.21	Parotid gland [13]
*CST5*	Cystatin D	20p11.21	Parotid gland, lacrimal gland [13]
*CST6*	Cystatin E/M	11q13.1	Epidermal tissue [15]
*CST7*	Cystatin F	20p11.21	Hematopoietic cells [16]
*CST8*	Cystatin 8	20p11.21	Testis [14]
*CST9*	Cystatin 9	20p11.21	Testis [14]
*CST11*	Cystatin 11	20p11.21	Testis [14]

**Table 2 cancers-15-05363-t002:** Summary of type 2 cystatins on their inflammation and tumor progression regulatory functions.

Name of Protein	Anti-Inflammatory Effect Pro-Tumor Effect	Pro-Inflammatory Effect Anti-Tumor Effect
CYS1	Promotes Th2 inflammatory response [85] Promotes M2 macrophage polarization [93] Inhibits JAK/STAT1 pathway [94] Inhibit tumor cell ferroptosis [123] Activates PI3K/AKT pathway [128] Partially neutralizes CYSC’s strong cathepsin B inhibitory effect [130]	Uncertain
CYS2	Uncertain May regulate the TGF-β signaling pathway [131]	Uncertain
CYSC	Suppresses MHC class II molecule maturation in immature DCs [105]	Stronger inhibitor of pro-tumor cathepsin B [147] Inhibits TGF- β pathway [149]
CYSS	Uncertain May promote EMT [134]	Uncertain
CYSD	Uncertain	Inhibits Wnt/β-catenin signaling pathway [139] Inhibits c-MYC expression [139]
CYS6	Suppress osteoclast differentiation and alleviate bone-matrix inflammatory stress [114]	Stronger inhibitor of pro-tumor legumain [160,161] Inhibits p38 activation induced by RANKL [162] Inhibits NF-κB signaling pathway [113,114] Suppresses osteoclast differentiation and prevents bone metastasis [114]
CYSF	Reduces NK cell and CD8+ T cell cytotoxicity [115,116] Regulates DC maturation [117]	Uncertain May be related to cathepsin B-L metastatic axis regulation

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
