# Peer review of "Type 2 Cystatins and Their Roles in the Regulation of Human Immune Response and Cancer Progression"

_cancers, 2023, doi:10.3390/cancers15225363_

Round 1

Reviewer 1 Report

Comments and Suggestions for Authors

In this review manuscript Zhang and Zhan aim to discuss the role of type 2 cystatins in the human immune response and in cancer. Although this is a very interesting topic, the manuscript often lacks depth (details) and includes blunt statements (leading to incorrect information). This should be corrected in order for the manuscript to be suitable for publication. Although a good comprehensive overview figure is made (Figure 3), the manuscript would really benefit from a table summarizing the contribution of each cystatin to the immune response/cancer and directly referring to the associated studies. Please find below my detailed comments:

Introduction, Page 1, lines 29-32: currently it sounds like all cathepsins are papain-like cysteine proteases. This is not the case. The statement that the human cathepsin family comprises 11 members is wrong. There are many more cathepsins that are not cysteine proteases.

Introduction, Page 1, lines 33-34: “whereas cathepsins K, S, W are solely expressed in onsteoclasts,…”. This statement is not correct! There are many more cell types that express these cathepsins.

General remark: please make sure that sufficient references are provided to all of the statements made. For example: lines 44-55 do not have any references. No references for lines 62-71,…

Table 1: where was this information obtained? References are missing?

Figures/tables: The manuscript would really benefit from a table summarizing the contribution of each cystatin to the immune response/cancer and directly referring to the associated studies.

Page 3: Please explain the name “SD-type cystatins”, what does SD stand for?

Page 3 + several other places: For several of the cystatins it is mentioned that these have antimicrobial activity. However, no details are provided about this. How does this work? Which pathogens? Which types of evidence are there (in vitro, in vivo,..)?

Page 5, lines 151-154: “Domain swapping properties of CYSC promote amyloid plaque formation,…”. This should be explained better. How does this work exactly? What is known about this?

Page 5, lines 172-174 + several other places: It is mentioned that the highest concentration of cystatin E/M is found in semen. However, no information is given on why this might be. It is also mentioned for other cystatins that they have a role in reproduction. This clearly shows that cystatins also have importance in reproduction and this should be better explained. Why? How? What?

Page 6, line 191: “In contrast, CYSF cannot inhibit cathepsin C and X in vitro?” this statement needs further explanation. Why could this be? What was the test system? Relavance?...

Page 6, line 193: “especially in immune cells” – please be more specific.

Page 6, lines 219-221: NSPs do not only have antimicrobial functions. They also modulate the immune response in several ways (see manuscript Pham 2006, Nat Rev Immunol - PMID: 16799473). This needs to be explained and nuanced better to avoid misinterpretation.

Page 6, lines 225-226: “overall, cathepsin C is a protease that plays an important role in immune cells to expert their cellular toxicity”. I don’t understand this. Please explain better.

Page 7, lines 251-253: these lines are not clear. Explain better.

Page 7, lines 263-264: “Cys1 serves as a biomarker of activated Th2 immune response”. Is there any information on which cell types this relates to? Which cells produce this?

Page 8, lines 291-293: this could be phrased better. It sounds as if the limitation in our knowledge is manely related to the biomarker field.

Page 8, lines 300-203: about coronavirus replication. More details are needed: in which assay/cell type/line?

Comments on the Quality of English Language

Language needs considerable revision, for example:

·       Title: Cystatins

·       The Simple Summary is poorly written (e.g. These enzymeS regulate..)

·       at times the manuscript is difficult to read and articles (e.g. "the") are missing

Author Response

To Reviewer 1:

Thank you so much for providing the feedback! I carefully checked each of them and will provide how I fixed these issues point by point in this document.

Also, I carefully revised the quality of English with a native speaker’s help and fixed the issue with the title. Please check whether the revised manuscript is easier to follow.

Introduction, Page 1, lines 29-32: currently it sounds like all cathepsins are papain-like cysteine proteases. This is not the case. The statement that the human cathepsin family comprises 11 members is wrong. There are many more cathepsins that are not cysteine proteases.

I fixed the statement about cathepsin. The fixed version talks about cysteine cathepsins specifically. Please check lines 29, 32 and 40.

Introduction, Page 1, lines 33-34: “Whereas cathepsins K, S, W are solely expressed in onsteoclasts,…”. This statement is not correct! There are many more cell types that express these cathepsins.

Thanks for the correction. Now this has been fixed to “predominantly expressed” (lines 33-34).

General remark: please make sure that sufficient references are provided to all of the statements made. For example: lines 44-55 do not have any references. No references for lines 62-71,…

Thank you for this advice. I added references in several spots in the fixed version including the place you mentioned.

Table 1: where was this information obtained? References are missing?

The gene location info was retrieved from the NCBI database and now it was addressed at line 85, references for the site of expression were also added.

Figures/tables: The manuscript would really benefit from a table summarizing the contribution of each cystatin to the immune response/cancer and directly referring to the associated studies.

Thanks for the advice. I added Table 2 (line 486) to summarize the contribution of each cystatin.

Page 3: Please explain the name “SD-type cystatins”, what does SD stand for?

The name “SD” comes from the combination of the 4 cystatins, “SN, SA, S, and D”. I added this info at line 88.

Page 3 + several other places: For several of the cystatins it is mentioned that these have antimicrobial activity. However, no details are provided about this. How does this work? Which pathogens? Which types of evidence are there (in vitroin vivo,..)?

The mechanism of the antimicrobial activity of cystatin remains unclear. But it could be that they can also inhibit growth-promoting proteolytic activity in bacteria. Most of the studies done on this are in vitro experiments using animal saliva. I also added several names of pathogens that were inhibited by cystatins. Please check lines 93-102.

Page 5, lines 151-154: “Domain swapping properties of CYSC promote amyloid plaque formation,…”. This should be explained better. How does this work exactly? What is known about this?

CYSC can form chain-like oligomers by its domain-swapping property and promote amyloid fibril formation. Please check lines 165-170 for a detailed explanation.

Page 5, lines 172-174 + several other places: It is mentioned that the highest concentration of cystatin E/M is found in semen. However, no information is given on why this might be. It is also mentioned for other cystatins that they have a role in reproduction. This clearly shows that cystatins also have importance in reproduction and this should be better explained. Why? How? What?

Very few papers have dug into this aspect. According to current existing literature, cystatin’s role in reproduction is indicated by its expression profile, but this role and the mechanism behind this is not fully understood. It has been reported that the loss of function of testatins is related to reduced fertility. Since this review is not focused on reproduction, I only added a very brief mention of this aspect. Please check lines 229-232.

Page 6, line 191: “In contrast, CYSF cannot inhibit cathepsin C and X in vitro?” this statement needs further explanation. Why could this be? What was the test system? Relavance?...

This phenomenon is caused by a post-translational modification. Cathepsin C cannot be inhibited by CYSF in the kinetic measurement experiment. But once the N-terminus region was cleaved, CYSF can inhibit cathepsin C, in both in vitro and in vivo experiments. Please see lines 209-214 for the detailed explanation.

Page 6, line 193: “especially in immune cells” – please be more specific.

I swapped “immune cells” into “NK cells, T cells, and DCs.” Please check line 214.

Page 6, lines 219-221: NSPs do not only have antimicrobial functions. They also modulate the immune response in several ways (see manuscript Pham 2006, Nat Rev Immunol - PMID: 16799473). This needs to be explained and nuanced better to avoid misinterpretation.

Thanks for this info! I added more info in this section mentioning their role in modifying cytokines, chemokines, and immune-related receptors. Please check lines 243-246.

Page 6, lines 225-226: “overall, cathepsin C is a protease that plays an important role in immune cells to expert their cellular toxicity”. I don’t understand this. Please explain better.

“Overall, cathepsin C activity is required in immune cells for exerting their cellular toxicity.” This fixed sentence is at line 250-251.

Page 7, lines 251-253: these lines are not clear. Explain better.

I rephrased these lines to “Since cystatins inhibit cathepsins and legumain, which in turn suppress inflammatory responses and immune cell activation, cystatins can be considered immune modulators.”

Please check lines 276-278.

Page 7, lines 263-264: “Cys1 serves as a biomarker of activated Th2 immune response”. Is there any information on which cell types this relates to? Which cells produce this?

CD4+ T cells are the main player in Th2 immune response. I added this info at line 290.

Page 8, lines 291-293: this could be phrased better. It sounds as if the limitation in our knowledge is manely related to the biomarker field.

This sentence is rephrased to “However, little is known about the other SD-type cystatins except that they can be used as biomarkers.”

This change is at lines 318-319

Page 8, lines 300-203: about coronavirus replication. More details are needed: in which assay/cell type/line?

It is done on MRC-5 diploid lung cells. Please check lines 327-330 for the answer.

Reviewer 2 Report

Comments and Suggestions for Authors

This is a very concise review about Type 2 cystatins and their implication in human immune response and cancer progression

I would suggest an organisational figure for the types of Cystatins

Comments on the Quality of English Language

Moderate editing of English language required

Author Response

Reviewer 2:

"This is a very concise review about Type 2 cystatins and their implication in human immune response and cancer progression

I would suggest an organisational figure for the types of Cystatins

Moderate editing of English language required"

To Reviewer 2:

Thanks for the suggestions! I added an organizational figure for the types of cystatins. Please check figure 1A at line 57.

Also, I carefully revised the quality of English with a native speaker’s help and fixed the issue with the title. Please check whether the revised manuscript is easier to follow.

Reviewer 3 Report

Comments and Suggestions for Authors

Major comments:

Part 3: CD4 T cells are not at all antigen presenting cells!

I am missing a part discussing the role and differences of the cystatin molecules in murine immunology; further a brief overview on available mouse models to study their biology would be helpful.

Minor comments:

Please take care about unneeded word doublings ('immune pathways and oncogenic pathways' should read 'immune and oncogenic pathways')

The quality of the figure 2 should be improved.

Comments on the Quality of English Language

no additional comments.

Author Response

To Reviewer 3:

Thanks for reviewing my manuscript! I have fixed most of the issues you mentioned. Please check my point-to-point feedback below.

Also, I carefully revised the quality of English with a native speaker’s help. Please check whether the revised manuscript is easier to follow.

Major comments:

Part 3: CD4 T cells are not at all antigen presenting cells!

I am sorry for this error. Since MHC-II maturation is a general phenomenon in APCs, I simply removed the example part in the fixed version. Please check lines 253-254.

I am missing a part discussing the role and differences of the cystatin molecules in murine immunology; further a brief overview on available mouse models to study their biology would be helpful.

I added a paragraph that talks about the difference in cystatins in mice in the conclusion. Overall, the mouse is a great tool for studying cystatins. Please check lines 510-521.

Minor comments:

Please take care about unneeded word doublings ('immune pathways and oncogenic pathways' should read 'immune and oncogenic pathways')

The quality of the figure 2 should be improved.

An improved version of Figure 2, showing only the key regions is now attached. At the same time, a higher resolution version of the original figure 2 is now the supplemental figure 1. Please check line 133 and line 524.

Round 2

Reviewer 3 Report

Comments and Suggestions for Authors

Thank you for the revised version.

Author Response

Thank you so much for signing the review report!

I wish you the best of luck in your future science life.